# Advanced Diffusion-Weighted Imaging Sequences for Breast MRI: Comprehensive Comparison of Improved Sequences and Ultra-High B-Values to Identify the Optimal Combination

**DOI:** 10.3390/diagnostics13040607

**Published:** 2023-02-07

**Authors:** Daniel Hausmann, Inga Todorski, Alexandra Pindur, Elisabeth Weiland, Thomas Benkert, Lars Bosshard, Michael Prummer, Rahel A. Kubik-Huch

**Affiliations:** 1Department of Radiology, Kantonsspital Baden, 5404 Baden, Switzerland; 2Department of Clinical Radiology and Nuclear Medicine, Medical Faculty Mannheim, Heidelberg University, 69117 Mannheim, Germany; 3Department of Radiology, Balgrist University Hospital, 8008 Zurich, Switzerland; 4MR Application Predevelopment, Siemens Healthcare GmbH, 91052 Erlangen, Germany; 5NEXUS Personalized Health Technologies, ETH Zurich, 8092 Zurich, Switzerland; 6Swiss Institute of Bioinformatics (SIB), 1015 Lausanne, Switzerland

**Keywords:** diffusion-weighted imaging, breast, noise reduction, magnetic resonance imaging, artifacts, image quality, breast cancer

## Abstract

This study investigated the image quality and choice of ultra-high b-value of two DWI breast-MRI research applications. The study cohort comprised 40 patients (20 malignant lesions). In addition to s-DWI with two m-b-values (b50 and b800) and three e-b-values (e-b1500, e-b2000, and e-b2500), z-DWI and IR m-b1500 DWI were applied. z-DWI was acquired with the same measured b-values and e-b-values as the standard sequence. For IR m-b1500 DWI, b50 and b1500 were measured, and e-b2000 and e-b2500 were mathematically extrapolated. Three readers used Likert scales to independently analyze all ultra-high b-values (b1500–b2500) for each DWI with regards to scan preference and image quality. ADC values were measured in all 20 lesions. z-DWI was the most preferred (54%), followed by IR m-b1500 DWI (46%). b1500 was significantly preferred over b2000 for z-DWI and IR m-b1500 DWI (*p* = 0.001 and *p* = 0.002, respectively). Lesion detection was not significantly different among sequences or b-values (*p* = 0.174). There were no significant differences in measured ADC values within lesions between s-DWI (ADC: 0.97 [±0.09] × 10^−3^ mm^2^/s) and z-DWI (ADC: 0.99 [±0.11] × 10^−3^ mm^2^/s; *p* = 1.000). However, there was a trend toward lower values in IR m-b1500 DWI (ADC: 0.80 [±0.06] × 10^−3^ mm^2^/s) than in s-DWI (*p* = 0.090) and z-DWI (*p* = 0.110). Overall, image quality was superior and there were fewer image artifacts when using the advanced sequences (z-DWI + IR m-b1500 DWI) compared with s-DWI. Considering scan preferences, we found that the optimal combination was z-DWI with a calculated b1500, especially regarding examination time.

## 1. Introduction

Breast MRI is the recommended method for tumor screening in high-risk patients, breast-implant evaluation, preoperative staging of breast cancer, and post-therapeutic follow-up [1,2,3,4]. In general, the dynamic contrast-enhanced (DCE) sequence with subtractions is the ideal sequence for detecting tumor angiogenesis. In the context of screening, annual controls are often performed in a young patient population, which leads to a high cumulative contrast agent load. Given the findings of gadolinium deposition in the brain, for which the clinical significance remains unclear, avoidance of repetitive contrast exposure is desirable [5,6]. In addition, a native protocol is of particular interest from a cost-efficiency perspective.

Diffusion-weighted imaging (DWI) is an established non-contrast method used for various body regions, such as the prostate, that indirectly detects tumor-typical cell-density increases by measuring Brownian molecular-motion restrictions in the intercellular space and allows the quantification of cellularity via the apparent diffusion coefficient (ADC) [7,8,9]. In breast MRI, DWI is used primarily to differentiate benign lesions from malignant lesions [10,11]. However, the value of DWI in the breast MRI protocol has long been questioned, especially due to the susceptibility of older sequences to artifacts. Recent technical achievements and new DWI techniques have significantly upgraded the value of DWI in breast imaging. A literature review published in 2022 indicates that DWI, diffusion tensor imaging (DTI), and their characteristics may facilitate an earlier and more accurate diagnosis, and consequent treatment improvements [12]. To ensure comparability of values, it is vital that values are not falsified by sequence properties. Moreover, it is particularly crucial that DWI can detect breast lesions independent of background enhancement, breast parenchymal density, and hormone status during the menstrual cycle or menopause [13,14]. The European Society of Breast Radiology recently published a consensus paper to promote and standardize the use of breast DWI in clinical practice [15].

At low b-values, T2-shine-through effects are often observed, whereas ultra-high b-values almost exclusively contain cell-density information and are therefore used primarily for cancer detection [16]. However, measuring high b-values is time-consuming, and image noise reduces image quality and distorts the calculated ADC value. A potential alternative is the extrapolation of ultra-high b-values without prolonging the scan time [17]. In the chest, sequences available to date are often affected by artifacts due to motion and inappropriate fat saturation (fs) in a relatively large field of view (FOV); thus, their use has not yet been established clinically. New zoomed techniques allow spatially tailored excitation pulses with a shortened echo train and reduced FOV in the phase-encoding direction and have already been shown to be beneficial in terms of image quality in the imaging of other organs [16,18,19] In addition, improvements in inversion-recovery (IR) fs may help optimize image quality, even at high measured b-values.

The aims of this study were: (1) to compare zoomed DWI with spectral-attenuated-IR (SPAIR) fs (z-DWI) and DWI with improved-IR fs (m-b1500 DWI) with the clinically established standard-DWI (s-DWI) sequence, and (2) to determine the optimal sequence and ultra-high b-value combination. ADC values were also calculated and compared between sequences. We hypothesized that new technical approaches would significantly improve the image quality of DWI data of the breast and thus promote the use of DWI for breast-cancer detection.

## 2. Materials and Methods

### 2.1. Study Population

After approval by the Ethics Committee of Northwestern and Central Switzerland (ID: 2020-00408), we reviewed all patients who had been registered for a breast MRI by in-house and external-referring physicians between August 2020 and June 2021. Exclusion criteria were patients aged < 18 years and those who had a contraindication to MRI examination, claustrophobia, or pregnancy. We also excluded patients who had undergone imaging for screening MRI and tumor follow-up, and those with breast implants. We included only patients who had undergone imaging on the 1.5 T MAGNETOM Aera scanner (Siemens Healthcare GmbH, Erlangen, Germany), on which the research application sequences were available. Of the patients registered for further MRI follow-up, we pre-selected patients who had been diagnosed with Breast Imaging-Reporting and Data System (BI-RADS) 4–6 lesions (BI-RADS 4: suspicious for malignancy; BI-RADS 5: highly suggestive of malignancy; BI-RADS 6: known biopsy-proven malignancy) by in-house or external physicians [20]. Preselecting patients with suspected or confirmed breast cancer (i.e., BI-RADS 4–6) ensured that we had a high proportion of malignant lesions that would allow meaningful evaluation of DWI data. On site, eligible patients were invited to participate voluntarily in the study after obtaining detailed informed consent.

### 2.2. MRI-Image Acquisition

In our institution, breast MRI is performed using a dedicated breast coil to avoid compressing patients’ breast tissue when they are positioned in the prone position, minimize artifacts, and optimize image quality [21]. According to international consensus, the current standard MRI protocol for breast imaging in the clinical setting comprises native T1- and T2-weighted image acquisition in the axial plane without fat suppression and s-DWI, followed by the application of 0.1 mmol contrast agent per kilogram of body weight for dynamic T1-weighted imaging [22].

### 2.3. DWI

Before contrast administration and immediately after the standard single-shot echo-planar imaging sequence (i.e., s-DWI), we acquired two study-specific advanced DWI sequences. We set up a zoomed single-shot echo-planar imaging sequence with a rotated field of excitation [23] (z-DWI sequence) using parameters comparable to the s-DWI sequence (spatial resolution 1.7 × 1.7 × 4.0 mm^3^, b-values 50/800 s/mm^2^), which provided the potential advantage of reduced aliasing and better image quality [24]. The m-b1500 DWI sequence was designed for higher b-value scanning by using short-tau inversion-recovery fs in combination with improved gradient reversal to avoid a residual-fat signal at the cost of an overall reduced signal. Therefore, a lower resolution was chosen (interpolated resolution 1.2 [i] × 1.2 [i] × 4.0 mm^3^, b-values 50/1500 s/mm^2^) and acquisition time was increased. All relevant protocol parameters are listed in Table 1.

### 2.4. Post Processing

For s-DWI and z-DWI, the b1500 value was calculated automatically at the MRI scanner console (Siemens MAGNETOM Aera, version Syngo MR E11, 91,052 Erlangen, Germany). In addition, b2000 and b2500 values were manually calculated using Syngo.via (VB40b_HF01 Siemens Healthcare GmbH 2009-2020, 91,052 Erlangen, Germany). Syngo.via allows the calculation of high b-values via a slider, with which the desired b-value can be selected. For IR-m1500 DWI, which measures b1500, the b2000 value was automatically calculated at the MRI console and the b2500 value was calculated manually, using Syngo.via.

### 2.5. Image Analysis

All diffusion-weighted images were independently reviewed by three radiologists (Reader A had over 20 years of experience in breast imaging; Reader B had over 10 years of experience in DWI and over 5 years of experience in breast imaging; and Reader C had 1 year of experience in breast imaging). The images were reviewed on a picture archiving and communication system (Centricity, GE Healthcare, Waukesha, WI, USA) workstation monitor (5 MP, Swiss BAG approved). The readers were informed only about the existence of a breast lesion before viewing the images, but were blinded to all other patient information. In cases in which a breast lesion was present, the readers could view the localization and size of the lesion/s on DCE T1-weighted images prior to the read-out. The three readers analyzed the images independently, using the following criteria.

Preferred sequence.Preferred b-value for each sequence (i.e., b1500, b2000, or b2500).

Criteria 3–5 for the four DWI sequences were rated on a Likert scale (1 = non-diagnostic; 2 = poor; 3 = fair; 4 = good; 5 = excellent). A value of 3 was assigned if the image impression, but not the interpretability of the sequence, was disturbed. Values of 1–2 indicated that artifacts or intracorporeal noise affected the interpretability or severely impacted the image impression (i.e., extracorporeal noise).

3.Extracorporeal noise in the proximity of the breast surface.4.Noise in the breast.5.Signal inhomogeneities and artifacts.

### 2.6. ADC Comparison

After subjective independent assessment of the image-quality characteristics, a region of interest with a similar size was carefully placed on the malignant lesions (only histopathologically confirmed lesions were used for the evaluation) on the ADC maps obtained from each sequence by consensus of all three readers, and the measured ADC values were compared between sequences.

### 2.7. Statistical Analysis

Statistical analysis was performed by Nexus at the Swiss Federal Institute of Technology, Zurich. All statistical analyses were performed using the R environment for statistical computing (R version 4.0.3 [10 October 2020]) and its dedicated packages. The analyses were conducted programmatically using R markdown 1 in Rstudio 2, which is compliant with the principles of reproducible research [25]. Chi-squared tests were used to determine significant differences between sequences, b-values, lesion-to-background contrast, extracorporeal noise, noise in the breast, and artifacts. Pairwise *t*-tests were used to compare ADC values between sequences. The significance level was set to 0.05, and Bonferroni correction was used for multiple comparisons. Cohen’s kappa was calculated to evaluate inter-reader reliability using the R package irr. The ratings for extracorporeal noise, noise in the breast, and artifacts were combined into three levels: 1 and 2 (bad); 3 (intermediate); 4 and 5 (good). This was to avoid poor agreement because of non-significant differences due to the generation of too many groups.

## 3. Results

### 3.1. Patients and Lesions

A total of 40 women were included (age: 59 ± 14 years). A malignant breast lesion was found and confirmed by biopsy in 20 patients, in whom false-negative findings were evaluated across all combinations of b-values and sequences. Lesion detection was not significantly different among sequences or b-values (*p* = 0.174). However, a trend toward a lower rate of false-negative findings was observed for the z-DWI sequence with a calculated b1500 value and the IR m-b1500 sequence with a measured b1500 value, compared with the standard protocol (Table 2).

### 3.2. Preferred Sequence

No reader preferred the standard sequence. The z-DWI sequence was the most preferred sequence (54%), followed by IR m-b1500 DWI (46%); however, there was no significant difference in preference between these two sequences overall (*p* = 0.36). We found that in the absence of a lesion, the z-DWI sequence was significantly preferred over the m-b1500 DWI sequence (*p* = 0.005), whereas in the presence of a lesion, the m-b1500 DWI sequence was preferred over the z-DWI sequence, albeit not significantly (*p* = 0.131; Figure 1).

### 3.3. Preferred B-Values

There were significant differences in preference between b-values (*p* < 0.001). Overall, the readers preferred b1500 (55%), followed by b2000 (44%), while b2500 was preferred for only 1% of cases. There was also a significant difference in preference between the most-preferred b-value, b1500, and the second-most-preferred b-value, b2000 (*p* = 0.017).

Figure 2 shows the preferred b-value distribution overall and for each sequence individually. Significant differences in the preference of b-value were obtained for all sequences (*p* < 0.001). B1500 was significantly preferred over b2000 for the z-DWI and IR m-b1500 DWI sequences (*p* = 0.001 and *p* = 0.002, respectively). However, b2000 was significantly preferred over b1500 for the standard sequence (*p* < 0.001; Figure 2).

### 3.4. Combination of B-Value and Sequence

The most preferred b-value and sequence combination was the z-DWI sequence with b1500; however, preference did not differ significantly from that of the IR m-b1500 DWI sequence (*p* = 0.48; Figure 3).

In the presence of a lesion, z-DWI with b1500 and IR m-b1500 with b1500 were preferred equally often (38.3%). There was no significant difference in preference between the two most-preferred b-value and sequence combinations (*p* = 1.00). In the absence of a lesion, z-DWI with b2000 was preferred most often (35%). There was no significant difference in preference between the most-preferred combination (i.e., z-DWI with b2000) and the second-most-preferred combination (z-DWI with b1500; *p* = 0.75). The distribution of preferences in the presence of a lesion and in patients without a lesion is shown in Table 3. In seven out of sixty cases (twenty patients with lesions and three readers), a lesion was not detected with the preferred sequence and b-value combination (z-DWI with b1500: *n* = 4; IR m-b1500 DWI: *n* = 3).

### 3.5. ADC Values

For the comparison of ADC values between the s-DWI, z-DWI, and IR m-b1500 DWI sequences, there was no significant difference between s-DWI (ADC: 0.97 [±0.09] × 10^−3^ mm^2^/s) and z-DWI (ADC: 0.99 [±0.11] × 10^−3^ mm^2^/s; *p* = 1.000). However, there was a non-significant trend toward lower values in the IR m-b1500 DWI sequence (ADC: 0.80 [±0.06] × 10^−3^ mm^2^/s) than in the s-DWI (*p* = 0.090) and z-DWI sequences (*p* = 0.110; Figure 4).

### 3.6. Noise in the Breast, Extracorporeal Noise, and Artifacts

There were significant differences between sequences in the extracorporeal noise, noise in the breast, and artifacts (*p* < 0.001). The z-DWI and IR m-b1500 DWI sequences generally achieved better results than the s-DWI sequence. IR m-b1500 DWI was rated significantly better than z-DWI across all three parameters (Figure 5).

### 3.7. Inter-Reader Agreement

There was good agreement between readers for artifacts. However, there was a low kappa value for extracorporeal noise and noise in the breast, which was primarily due to disagreement between the intermediate and good ratings (Table 4).

## 4. Discussion

We demonstrated that the advanced DWI sequences had an advantage over the standard clinical sequence across all image-quality characteristics. The overall preferred b-value was b1500, and the most-preferred b-value and sequence combination was the z-DWI sequence, with an extrapolated b-value of b1500. The present-study findings may contribute to the selection of appropriate sequence-protocol parameters and thus further improve the clinical significance of DWI in breast MRI.

DWI has the potential to become a standalone technique, rather than a supplementary method, for decision-making, to distinguish potentially malignant lesions from benign lesions in the breast. Moreover, in the medium-to-long term, it is anticipated that intravenous administration of contrast agents will become unnecessary [26]. DWI can detect breast lesions independently of background enhancement, breast parenchymal density, and hormone status during the menstrual cycle or menopause [13], which simplifies scheduling in clinical practice. For screening and specialized incompatible cases, it has even been postulated that DCE-MRI may be omitted, and precise diagnostic accuracy can be achieved using DWI-MRI [27]. Furthermore, DWI sequences are important for not only confirming lesions and distinguishing malignant lesions from benign lesions, but also for achieving the medium-term goal of reducing or even eliminating the need for intravenous contrast-agent administration while maintaining high diagnostic accuracy [28,29]. To enable international comparability in diagnostics by standardizing MRI-breast-examination methods and techniques, the development of new, clinically relevant, and validated techniques by consensus is crucial [1,30].

In DWI, selecting the correct b-value is crucial, because the b-value influences the signal-to-noise ratio, contrast-to-noise ratio (CNR), and ADC value [31]. Ohlmeyer et al. demonstrated that increasing the b-value and using ultra-high b-values enable more precise evaluations because tissue with higher diffusivity, such as normal fibroglandular tissue, can be further suppressed [32]. Tamura et al. demonstrated that there is an upper limit for the diffusion value, where the peak of the CNR is exceeded when values above 2500 s/mm^2^ are used [33]. Thus, overall, very high b-values between 1500 and 2500 s/mm^2^ are recommended for diagnostic precision. In our study, the measured and calculated b1500 values were preferred over b2000 and b2500, and at higher b-values in particular, lesions were often more difficult to delineate, although there was no effect on the lesion- detection rate (Figure 3). In our view, therefore, b1500 should not be exceeded. In the case of b-value measurement, this would also have advantages with regard to the time and thus cost efficiency of the DWI, because b1500 requires fewer averages and thus a shorter measurement time compared with higher b-values.

The primary reason that the use of DWI for diagnostic breast MRI for clinical decision-making is controversial, despite the well-established use of standalone DWI for other body regions, is the technical inconsistency of image quality because of the artifact susceptibility of DWI sequences [15,34,35] Recently, an increasing number of studies have been investigating not only new approaches for basic DWI sequences for breast imaging but also other parameters, such as accelerating the acquisition of DWI sequences to further reduce motion artifacts. Biswas et al. published a prospective study in 2022 that investigated the clinical value of accelerated multiband sensitivity-encoding (MB SENSE) DWI in 38 women [26]. No differences were observed between MB SENSE DWI and conventional DWI (cDWI) in ADC measurements (*p* = 0.50), CNR (*p* = 0.17), or signal intensity (*p* = 0.23). However, the image quality of cDWI and MB SENSE DWI was considered equivalent in 51% of images, and MB SENSE DWI was preferred more often than cDWI (*p* < 0.001). The preference for MB SENSE DWI over cDWI was primarily attributed to better fat suppression.

The use of z-DWI techniques with spatially tailored excitation pulses and reduced FOV in the phase-encoding direction has already been proven successful in organs other than the breast, owing to improved image quality and reduced distortion and susceptibility artifacts [16,18,19,36]. Previous studies have mostly used zoomed diffusion with spatially tailored excitation pulses through parallel transmit. The 1-channel z-DWI applied in the present study uses a rotated field of excitation to reduce the echo time and artifacts such as aliasing. In our study, z-DWI had fewer artifacts and less noise compared with s-DWI. In particular, extracorporeal-noise reduction contributed to an improved image impression. However, this did not significantly impact diagnostic performance.

Another common challenge of breast DWI is artifacts in the posterior portion of the breast due to a residual-fat signal (Figure 3). The IR m-b1500 DWI used in the current study, unlike the other two techniques investigated, does not use SPAIR fat saturation, but IR fat saturation, which should help to reduce the residual-fat signal and thus enable measurement of ultra-high b-values with improved image quality. In a recent study using a 3T scanner, fat suppression was shown to impact ADC calculation in benign and malignant breast lesions (*p* = 0.013 and *p* = 0.001, respectively) [37], which is problematic when using ADC values to predict tumor grade [16]. Another important variable is the method by which high b-values are generated. DelPriore et al. published a study in 2022 that indicated that measured high b-values have a slight advantage over calculated b-values, in terms of CNR [38].

In our study, the improved fat suppression of the IR m-b1500 DWI sequence had a positive effect on the image impression compared with the s-DWI sequence with SPAIR fs (Figure 3). IR m-b1500 DWI achieved the best results for the image-quality parameters of extracorporeal noise, noise in the breast, and artifacts. However, when a lesion was present, IR m-b1500 DWI and z-DWI with b1500 were equally often preferred, and compared with the other sequences, the acquisition time of IR m-b1500 DWI was increased by almost 1 min. However, noise at measured high b-values can result in falsely low ADC values. In malignant lesions, ADC values calculated from b50 and b1500 tended to be lower than those calculated from b50 and b800 using the s-DWI and z-DWI sequences (Figure 4), respectively, and it is unclear whether the ADC value was affected by noise due to the high b-value or due to the IR fat saturation. Nevertheless, the results indicate that ADC values should be calculated without the measured b1500 to secure comparability of ADC values and to predict tumor grade. However, the measurement of an additional b-value (e.g., b800) would result in prolonged acquisition time.

### Limitations

The study has several major limitations. First, the study was conducted in a small heterogeneous population of women with and without breast lesions, and the sample size of women with breast lesions was insufficient to draw generalizable conclusions. Second, with the exception of the ADC-value comparison, the analyses were subjective. Third, the studies were only conducted on a 1.5 T MRI scanner, which limits the generalizability of our results to other scanner strengths.

## 5. Conclusions

Overall, we observed better image quality and fewer image artifacts across all b-values using advanced sequences (z-DWI and IR m-b1500 DWI) compared with s-DWI. However, there was no significant improvement in lesion detectability in our small dataset. Furthermore, a b-value of 1500 appears to offer the best compromise in terms of image quality, lesion detectability, and measurement time. The IR fs allows the measurement of high b-values (b1500) with very good image quality by eliminating the residual-fat signal. Targeted studies are required to determine whether the ADC value in malignant lesions is actually confounded by the measurement (either by the nature of fs or by visually undetectable noise).

## Figures and Tables

**Figure 1 diagnostics-13-00607-f001:**
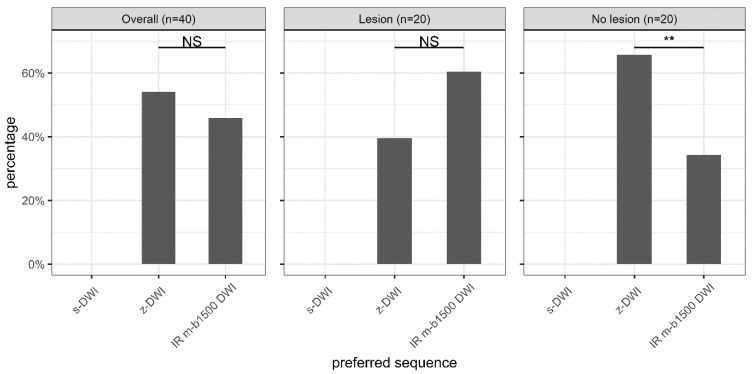
Zoomed diffusion-weighted imaging (DWI) with spectral-attenuated inversion-recovery fat saturation (z-DWI) was the preferred sequence overall, although not significantly. Significant differences in preference were observed only in the absence of a lesion, where DWI with improved inversion-recovery fat saturation (IR m-b1500 DWI) was preferred. s-DWI: single-shot echo-planar DWI. NS = *p* > 0.05. ** = 0.001 < *p* ≤ 0.01.

**Figure 2 diagnostics-13-00607-f002:**
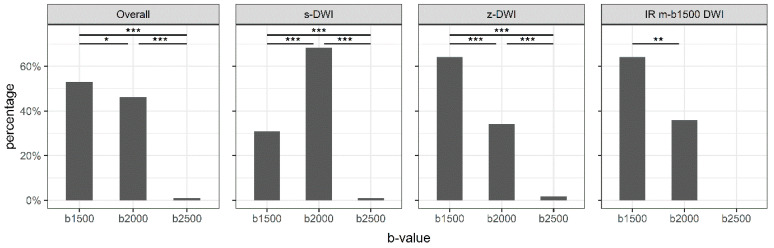
Preferred b-values for the different sequences. The preferred b-value is dependent on the sequence. s-DWI: single-shot echo-planar DWI; z-DWI: zoomed DWI with spectral-attenuated inversion-recovery fat saturation; IR m-b1500 DWI: DWI with improved inversion-recovery fat saturation. * = 0.01 < *p* ≤ 0.05; ** = 0.001< *p* ≤ 0.01; *** = *p* ≤ 0.001.

**Figure 3 diagnostics-13-00607-f003:**
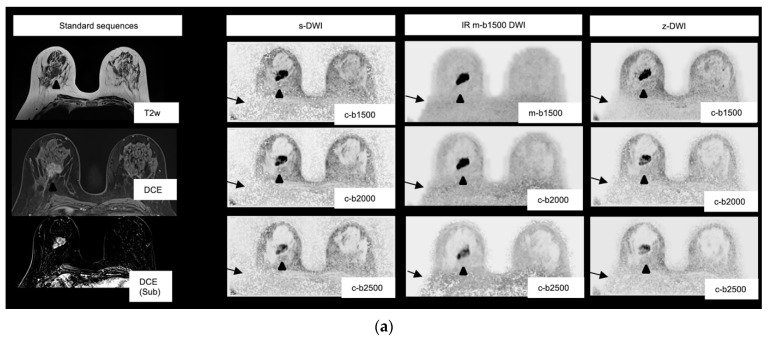
(**a**) Images of a 65-year-old patient with a 3.2 cm BIRADS (Breast Imaging Reporting and Data System) 5 lesion in the right breast in the 8–9 o’clock axis. Histopathology showed a ductal breast carcinoma (G3). The lesion is well-demarcated for all sequences and high b-values (arrowheads). Overall, the black and white inverted m-b1500 sequence with a measured b1500 was preferred. The signal inhomogeneities in the posterior part of the right breast are particularly reduced in the improved inversion-recovery fat-saturation diffusion-weighted-imaging (IR m-b1500 DWI) images (arrows). s-DWI: single-shot echo-planar DWI; z-DWI: zoomed DWI with spectral-attenuated inversion-recovery fat saturation. (**b**) Images of an 84-year-old patient with a 1.6 cm lesion in the left breast in the 11 o’clock axis. Histopathology showed a ductal breast carcinoma (G2). Noise and artifacts were rated better for the IR m-b1500 and zoomed DWI with spectral-attenuated inversion-recovery fat-saturation (z-DWI) images than for the single-shot echo-planar DWI (s-DWI) images. Again, signal inhomogeneities in the posterior part of the right breast are particularly reduced in IR m-b1500 DWI (IR m-b1500 DWI) (arrows). The lesion is most visible in the b1500 images compared with the b2000/b2500 images, owing to superior lesion-to-background contrast (arrowheads).

**Figure 4 diagnostics-13-00607-f004:**
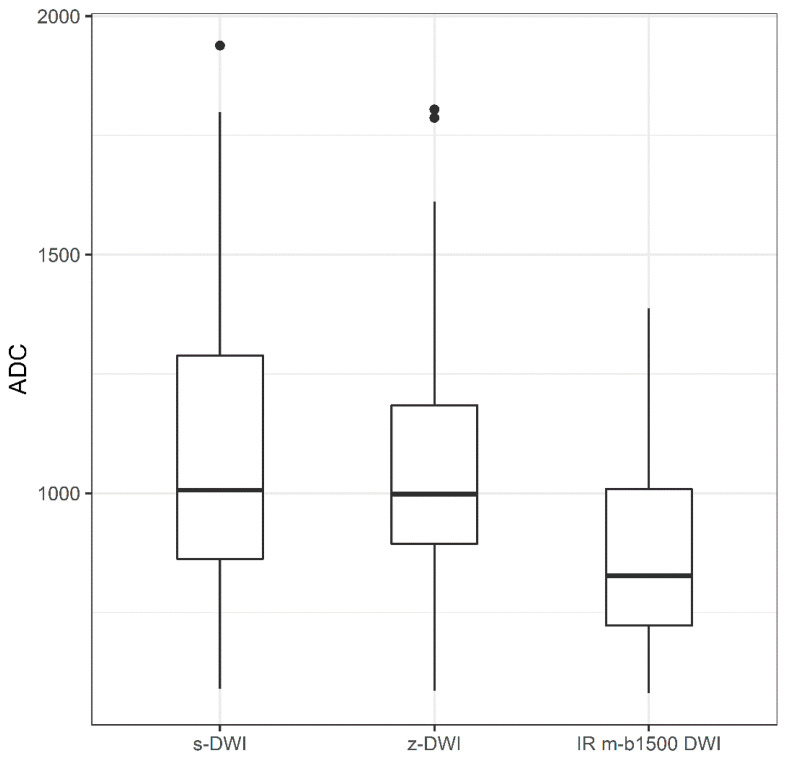
Boxplot of apparent-diffusion-coefficient (ADC) values of the single-shot echo-planar diffusion-weighted imaging (DWI) (s-DWI), zoomed DWI with spectral-attenuated inversion-recovery fat saturation (z-DWI), and improved inversion-recovery fat-saturation DWI (IR m-b1500 DWI) sequences.

**Figure 5 diagnostics-13-00607-f005:**
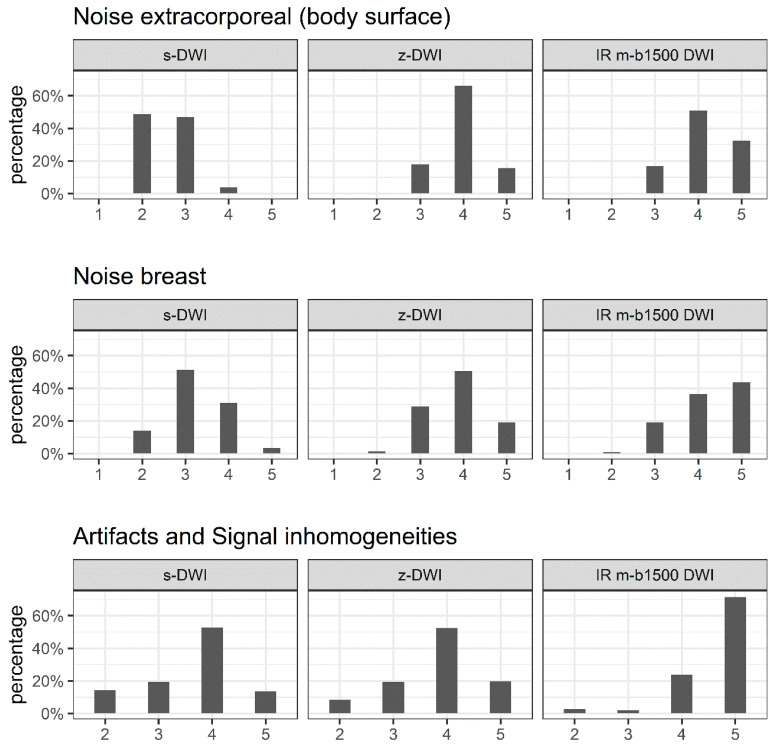
Distribution of ratings for extracorporeal noise, noise in the breast, and artifacts scored on a Likert scale (1 = poor; 2 = fair; 3 = good; 4 = very good; 5 = excellent). We tested for significant differences in Likert scores between the three sequences (single-shot echo-planar diffusion-weighted imaging (DWI) (s-DWI)), zoomed DWI with spectral-attenuated inversion-recovery fat saturation (z-DWI), and improved inversion-recovery fat-saturation DWI (IR m-b1500 DWI) for each variable. Extracorporeal noise: *p* < 0.001 for s-DWI vs. z-DWI, s-DWI vs. IR m-b1500 DWI, and z-DWI vs. IR m-b1500 DWI. Noise in the breast: *p* < 0.001 for s-DWI vs. z-DWI, s-DWI vs. IR m-b1500 DWI, and z-DWI vs. IR m-b1500 DWI. Artifacts: *p* = 0.024 for s-DWI vs. z-DWI, *p* < 0.001 for s-DWI vs. IR m-b1500 DWI and z-DWI vs. IR m-b1500 DWI.

**Table 1 diagnostics-13-00607-t001:** Diffusion-weighted-imaging (DWI) protocol parameters.

Parameters	s-DWI	z-DWI	IR m-b1500 DWI
Measured b-values (s/mm^2^)	50; 800	50; 800	50; 1500
Calculated b-values (s/mm^2^)	1500; 2000; 2500	1500; 2000; 2500	2000; 2500
Voxel size (mm^3^)	1.7 × 1.7 × 4.0	1.7 × 1.7 × 4.0	1.2 (i) × 1.2 (i) × 4.0
Field of view (mm^2^)	163 × 340	163 × 340	212 × 340
Fat saturation	SPAIR	SPAIR	IR
Acquisition time (min)	2:34	2:23	3:22
Echo time (ms)	59	63	64
Repetition time (ms)	5300	5300	7560

s-DWI: single-shot echo-planar DWI sequence; z-DWI: zoomed DWI with spectral-attenuated inversion-recovery fat saturation; IR m-b1500 DWI: DWI with improved inversion-recovery fs and measured b1500; SPAIR: spectral-attenuated inversion recovery; IR: inversion recovery.

**Table 2 diagnostics-13-00607-t002:** Percentage of undetected lesions for each sequence and b-value combination.

Sequence	b1500	b2000	b2500
z-DWI	6.1%	13.6%	18.4%
IR m-b1500 DWI	4.1%	6.8%	11.6%
s-DWI	8.2%	13.6%	17.7%

DWI: diffusion-weighted imaging; s-DWI: single-shot echo-planar DWI sequence; z-DWI: zoomed DWI with spectral-attenuated inversion-recovery fat saturation; IR m-b1500 DWI: DWI with improved inversion-recovery fat saturation and measured b1500.

**Table 3 diagnostics-13-00607-t003:** Sequence and b-value preference (number of selections by each reader and the percentage of selections across all readers).

(a) With a Lesion				
Reader	Sequence	b1500	b2000	b2500
Reader A	z-DWI	10	0	0
	IR m-b1500 DWI	7	3	0
Reader B	z-DWI	7	2	0
	IR m-b1500 DWI	7	4	0
Reader C	z-DWI	6	0	0
	IR m-b1500 DWI	9	5	0
Overall	z-DWI	38.3%	3.3%	0
	IR m-b1500 DWI	38.3%	20%	0
**(b) Without a Lesion**				
**Reader**	**Sequence**	**b1500**	**b2000**	**b2500**
Reader A	z-DWI	13	0	0
	IR m-b1500 DWI	6	1	0
Reader B	z-DWI	1	15	1
	IR m-b1500 DWI	3	0	0
Reader C	z-DWI	4	6	0
	IR m-b1500 DWI	7	3	0
Overall	z-DWI	30%	35%	1.7%
	IR m-b1500 DWI	26.7%	6.7%	0

DWI: diffusion-weighted imaging; s-DWI: single-shot echo-planar DWI; z-DWI: zoomed DWI with spectral-attenuated inversion-recovery fat saturation; IR m-b1500 DWI: DWI with improved inversion-recovery fat saturation and measured b1500.

**Table 4 diagnostics-13-00607-t004:** Pairwise reader reliability (Cohen’s kappa) for each sequence. A kappa value of > 0.21 was considered sufficient (marked in bold).

s-DWI	Reader A vs. B	Reader A vs. C	Reader B vs. C
Extracorporeal noise	0.071	0.022	0.032
Noise in the breast	0.072	0.112	0.037
Artifacts	**0.547**	**0.375**	**0.554**
**z-DWI**	**Reader A vs. B**	**Reader A vs. C**	**Reader B vs. C**
Extracorporeal noise	0.072	**0.282**	0.018
Noise in the breast	0.031	**0.382**	0.005
Artifacts	**0.512**	**0.309**	**0.475**
**m-b1500 DWI**	**Reader A vs. B**	**Reader A vs. C**	**Reader B vs. C**
Extracorporeal noise	0.070	**0.559**	**0.236**
Noise in the breast	**0.386**	**0.406**	**0.324**
Artifacts	**0.719**	**0.587**	**0.519**

DWI: diffusion-weighted imaging; s-DWI: single-shot echo-planar DWI; z-DWI: zoomed DWI with spectral-attenuated inversion-recovery fat saturation; IR m-b1500 DWI: DWI with improved inversion-recovery fat saturation and measured b1500.

## Data Availability

Anonymized patient data could be provided on request.

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
