# Peer review of "Advanced Diffusion-Weighted Imaging Sequences for Breast MRI: Comprehensive Comparison of Improved Sequences and Ultra-High B-Values to Identify the Optimal Combination"

_diagnostics, 2023, doi:10.3390/diagnostics13040607_

Round 1

Reviewer 1 Report

The topic is of interest, and the manuscript is well illustrated.

Major Comments:

1. Are there controversies in this field? What are the most recent and important achievements in the field? In my opinion, answers to these questions should be emphasized. Perhaps, in some cases, novelty of the recent achievements should be highlighted by indicating the year of publication in the text of the manuscript.

2. The results and discussion section is very weak and no emphasis is given on the discussion of the results like why certain effects are coming in to existence and what could be the possible reason behind them?

3. Conclusion: not properly written.

4. Results and conclusion: The section devoted to the explanation of the results suffers from the same problems revealed so far. Your storyline in the results section (and conclusion) is hard to follow. Moreover, the conclusions reached are really far from what one can infer from the empirical results.

5. The discussion should be rather organized around arguments avoiding simply describing details without providing much meaning. A real discussion should also link the findings of the study to theory and/or literature.

6. Spacing, punctuation marks, grammar, and spelling errors should be reviewed thoroughly. I found so many typos throughout the manuscript.

7. English is modest. Therefore, the authors need to improve their writing style. In addition, the whole manuscript needs to be checked by native English speakers.

Author Response

Dear Reviewer,

thank you for your review and the insightful critique that helped us improve the quality of our manuscript.

Point 1:  Are there controversies in this field? What are the most recent and important achievements in the field? In my opinion, answers to these questions should be emphasized. Perhaps, in some cases, novelty of the recent achievements should be highlighted by indicating the year of publication in the text of the manuscript.

Thank you for the valuable comment. For a long time, the value of DWI in the breast MRI protocol was questioned, especially due to the artifact susceptibility of older sequences. Recent technical achievements and new DWI techniques have significantly upgraded the value of DWI in breast imaging.

A recent Diagnostics literature overview was cited in the Introduction section (page 2):

"However, the value of DWI in the breast MRI protocol has long been questioned, especially due to the susceptibility of older sequences to artifacts. Recent technical achievements and new DWI techniques have significantly upgraded the value of DWI in breast imaging. A literature review published in 2022 indicates that DWI, diffusion tensor imaging (DTI), and their characteristics may facilitate an earlier and more accurate diagnosis, and consequent treatment improvements.12"

Point 2: The results and discussion section is very weak and no emphasis is given on the discussion of the results like why certain effects are coming in to existence and what could be the possible reason behind them?

In the results section, we have tried to compile and illustrate the results as comprehensively as possible. I agree with you that in the discussion section we do not sufficiently refer to the results and put them into the scientific context.

The following addition has been made to the Discussion section:

Page 11:

"The present study findings may contribute to the selection of appropriate sequence protocol parameters and thus further improve the clinical significance of DWI in breast MRI."

Page 11:

"In our view, therefore, a b1500 should not be exceeded. In the case of b-value measurement, this would also have advantages with regard to the time and thus cost efficiency of the DWI, because with a b1500, fewer averages and thus a shorter measurement time are necessary compared to the higher b-values."

Point 3: Conclusion: not properly written.

The conclusion section was completely written according to your comment:

Page 12:

" Overall, we observed better image quality and fewer image artifacts across all b-values using advanced sequences (z-DWI and IR m-b1500 DWI) compared with s-DWI. However, there was no significant improvement in lesion detectability in our small da-taset. Furthermore, a b-value of 1500 appears to offer the best compromise in terms of image quality, lesion detectability, and measurement time. The IR fs allows the meas-urement of high b-values (b1500) with very good image quality by eliminating the re-sidual fat signal. Targeted studies are required to determine whether the ADC value in malignant lesions is actually confounded by the measurement (either by the nature of fs or by visually undetectable noise)."

Point 4: Results and conclusion: The section devoted to the explanation of the results suffers from the same problems revealed so far. Your storyline in the results section (and conclusion) is hard to follow. Moreover, the conclusions reached are really far from what one can infer from the empirical results.

Please refer to Point 3+4.

Point 5: The discussion should be rather organized around arguments avoiding simply describing details without providing much meaning. A real discussion should also link the findings of the study to theory and/or literature.

Please refer to Point 3+4.

Point 6: Spacing, punctuation marks, grammar, and spelling errors should be reviewed thoroughly. I found so many typos throughout the manuscript.

Thank you for the comment. The manuscript was reviewed by EDANZ before submission. We have now had the manuscript linguistically revised again by EDANZ. In the future, we will use the editing service of Diagnostics.

Point 7: English is modest. Therefore, the authors need to improve their writing style. In addition, the whole manuscript needs to be checked by native English speakers.

Please see above.

Reviewer 2 Report

The article by Daniel Hausmann et al. entitled “Advanced diffusion-weighted imaging sequences for breast MRI: Comprehensive comparison of improved sequences and ultra-high b-values – what is the perfect match?” is quite interesting. But it raises the following issue.

1.  Author must write the aim of the study in the abstract
2. The abstract should be no more than 200 words long. The abstract should be a single paragraph and should follow the style of structured abstracts, but without headings. The author must rewrite the abstract according to the author's instructions.
3.  Abbreviations should be written after the conclusion
4.  In the text, reference numbers should be placed in square brackets [ ], and placed before the punctuation; for example [1], [1–3], or [1,3].
5. Discussion: Authors should discuss the results and how they can be interpreted from the perspective of previous studies and the working hypotheses. Future research directions may also be mentioned. 
6. The reference list needs to be rewritten and should include the author list, the full title, journal name,  volume, and page number as recommended by the ACS style guide or according to the author's instructions.
7. Minor typographical errors were found throughout the manuscript and should be corrected.

8. Overall, the authors need to rewrite the manuscript according to the journal format

Author Response

Dear Reviewer,

thank you for your review and the insightful critique that helped us improve the quality of our manuscript.

Point 1:  Author must write the aim of the study in the abstract

Thank you for your comment.

The aim was added a the beginning of the abstract:

Page 1:

"This study investigated the image quality and choice of ultra-high b-value of two DWI breast MRI research applications."

  1. The abstract should be no more than 200 words long. The abstract should be a single paragraph and should follow the style of structured abstracts, but without headings. The author must rewrite the abstract according to the author's instructions.

The abstract has been significantly shortened and combined into one block, but still slightly exceeds the 200 word limit. We are happy to make an additional shortening if you wish.

  1. Abbreviations should be written after the conclusion

The explanation of abbreviations has been moved to the Abbreviations list immediately after the Conclusion

  1. In the text, reference numbers should be placed in square brackets [ ], and placed before the punctuation; for example [1], [1–3], or [1,3].

The manuscript was revised according to your comment.

  1. Discussion: Authors should discuss the results and how they can be interpreted from the perspective of previous studies and the working hypotheses. Future research directions may also be mentioned. 

The following addition has been made to the Discussion section:

Page 11:

"The present study findings may contribute to the selection of appropriate sequence protocol parameters and thus further improve the clinical significance of DWI in breast MRI."

Page 11:

"In our view, therefore, a b1500 should not be exceeded. In the case of b-value measurement, this would also have advantages with regard to the time and thus cost efficiency of the DWI, because with a b1500, fewer averages and thus a shorter measurement time are necessary compared to the higher b-values."

The conclusion section was completely written according to your comment:

Page 12:

" Overall, we observed better image quality and fewer image artifacts across all b-values using advanced sequences (z-DWI and IR m-b1500 DWI) compared with s-DWI. However, there was no significant improvement in lesion detectability in our small da-taset. Furthermore, a b-value of 1500 appears to offer the best compromise in terms of image quality, lesion detectability, and measurement time. The IR fs allows the meas-urement of high b-values (b1500) with very good image quality by eliminating the re-sidual fat signal. Targeted studies are required to determine whether the ADC value in malignant lesions is actually confounded by the measurement (either by the nature of fs or by visually undetectable noise)."

  1. The reference list needs to be rewritten and should include the author list, the full title, journal name, volume, and page number as recommended by the ACS style guide or according to the author's instructions.

The manuscript was revised according to your comment.

  1. Minor typographical errors were found throughout the manuscript and should be corrected.

Thank you for the comment. The manuscript was reviewed by EDANZ before submission. We have now had the manuscript linguistically revised again by EDANZ. In the future, we will use the editing service of Diagnostics.

  1. Overall, the authors need to rewrite the manuscript according to the journal format

Please refer to point 7.

Reviewer 3 Report

Interesting data. I have no specific comments

Author Response

Thank you very much. We are very pleased with the positive review.

The manuscript was reviewed by EDANZ before submission. We have now had the manuscript linguistically revised again by EDANZ. In the future, we will use the editing service of Diagnostics.

Round 2

Reviewer 1 Report

The discussion should be rather organized around arguments avoiding simply describing details without providing much meaning. A real discussion should also link the findings of the study to theory and/or literature.

Author Response

Response to Reviewer 1 Comments

Dear Reviewer,

Thank you for the additional comment.

Point 1:  The discussion should be rather organized around arguments avoiding simply describing details without providing much meaning. A real discussion should also link the findings of the study to theory and/or literature.

We have attempted to discuss the technical rationale of the z-DWI and IR m-b1500 DWI in the context of the current literature and to highlight differences from old techniques.

In the discussion, we made the following additions:

Page 12:

"Previous studies have mostly used zoomed diffusion with spatially tailored excitation pulses through parallel transmit. The 1-channel z-DWI applied in the present study uses a rotated field-of-excitation to reduce the echo time and artifacts such as aliasing.."

Page 12:

"Another common challenge of breast DWI is artifacts in the posterior portion of the breast due to residual fat signal (Figure 3).  The IR m-b1500 DWI used in the current study, unlike the other two techniques investigated, does not use SPAIR fat saturation, but IR fat saturation, which should help to reduce residual fat signal and thus enable measurement of ultra-high b-values with improved image quality."

Reviewer 2 Report

The present form is accepted for publication.

Author Response

Dear Reviewer,

thank you very much for your review. We are happy that we were able to improve the manuscript through their comments.

Thank you and kind regards!